# Invasion Dynamics and Migration Patterns of Fall Armyworm (*Spodoptera frugiperda*) in Shaanxi, China

**DOI:** 10.3390/insects16060620

**Published:** 2025-06-11

**Authors:** Zhanfeng Yan, Xiaojun Feng, Xing Wang, Xiangqun Yuan, Yongjun Zhang, Daibin Yang, Kanglai He, Feizhou Xie, Zhenying Wang, Yiping Li

**Affiliations:** 1Key Laboratory of Plant Protection Resources and Pest Management of Ministry of Education, Key Laboratory of Integrated Pest Management on Crops in Northwestern Loess Plateau of Ministry of Agriculture and Rural Affairs, College of Plant Protection, Northwest A&F University, Yangling 712100, China; yanmaurice@163.com (Z.Y.); 17629086883@163.com (X.W.); yuanxq@nwsuaf.edu.cn (X.Y.); 2State Key Laboratory of Crop Germplasm Innovation and Molecular Breeding, Syngenta Biotechnology (China) Co., Ltd., Beijing 102206, China; 3Plant Protection Station of Shaanxi Province, Xi’an 710003, Chinaxiefeizhou@126.com (F.X.); 4State Key Laboratory for Biology of Plant Diseases and Insect Pests, Institute of Plant Protection, Chinese Academy of Agricultural Sciences, Beijing 100193, China; yjzhang@ippcaas.cn (Y.Z.); yangdaibin@caas.cn (D.Y.); klhe@ippcaas.cn (K.H.)

**Keywords:** invasive pest, migration route, molecular characterization, overwinter

## Abstract

The fall armyworm (FAW) is a serious invasive pest that has caused significant damage to crops in China since its arrival in 2019. This study investigates how FAW spreads and migrates in Shaanxi Province over five years, using field surveys, computer simulations, genetic analyses, and experiments. We found that FAW’s appearance in Shaanxi has been delayed each year, and its migration routes and source regions vary annually. The main sources of FAW in Shaanxi are from neighboring provinces like Sichuan, Chongqing, Hubei, Guizhou, and Henan. Genetic analysis showed that FAW populations in Shaanxi have high diversity, with different strains mixing together. Overwintering experiments revealed that FAW pupae cannot survive the winter in Shaanxi, indicating that the region cannot support year-round breeding of this pest. Our findings help us understand FAW’s behavior and provide valuable information for developing better pest management strategies to protect crops and food security in China and beyond.

## 1. Introduction

The fall armyworm (FAW), *Spodoptera frugiperda*, is a highly significant pest that poses a considerable threat to global food security, particularly in regions where maize cultivation is prevalent [1,2]. Since its initial detection in Africa in 2016 and its subsequent invasion into China in 2019, understanding the migration ecology of FAW has become crucial for developing science-based management strategies [3]. Monitoring insect migration is essential for predicting and managing pest outbreaks. As Johnson [4] first comprehensively described, insect migration is a complex phenomenon influenced by various factors, including environmental cues and physiological states. For highly mobile pests like FAW, effective monitoring of their long-distance movements is vital for timely interventions and sustainable management practices [5,6]. The importance of migration monitoring in FAW management cannot be overstated. Early detection through monitoring systems enables rapid responses, potentially preventing widespread crop damage and reducing the need for extensive pesticide use [7,8]. Furthermore, understanding migration patterns aids in predicting future infestation areas, allowing for proactive rather than reactive management strategies [9].

In China, FAW has exhibited two primary migration routes: the western route, originating in Myanmar and Yunnan and passing through Guizhou, Chongqing, Sichuan, and Shaanxi, and the eastern route, starting from northern Thailand, Laos, Vietnam, Guangxi, and Guangdong and proceeding through southeastern and east-central provinces to the Huang-Huai-Hai and Northeast Regions [10,11]. However, a critical gap remains in our understanding of FAW’s invasion dynamics, particularly in transitional climatic zones. Previous studies have mainly focused on the general migration patterns and genetic characteristics of FAW in China. Yet, the complex interactions between migration routes, genetic diversity, and overwintering ability in specific transitional regions have not been adequately addressed. Additionally, the potential for genetic exchange between the eastern and western routes and its implications for FAW’s adaptability and management strategies remain underexplored.

This study hypothesizes that the unique migration patterns and genetic diversity of FAW in Shaanxi contribute to its invasive success and that the region’s climate significantly limits its overwintering ability. To fill these knowledge gaps, this study comprehensively investigates FAW’s invasion dynamics in Shaanxi Province from 2019 to 2023. By integrating field surveys, trajectory simulations, molecular analysis, and overwintering experiments, we aim to achieve the following objectives: (1) determine the annual variation in FAW occurrence and identify its migration routes and source regions; (2) assess the genetic diversity and strain composition of FAW populations in Shaanxi and surrounding areas; and (3) evaluate FAW’s overwintering ability in Shaanxi and its implications for regional pest management. Our research not only enhances the understanding of FAW’s behavior in a critical transitional zone but also challenges existing assumptions by revealing the complexity of population mixing and genetic exchange. These findings provide valuable insights for refining pest management strategies and improving early warning systems to protect agricultural productivity and food security in China and beyond.

## 2. Materials and Methods

### 2.1. Survey of FAW Initial Infestation in Shaanxi

This study was conducted in six cities of Shaanxi, Hanzhong, Ankang, Shangluo, Baoji, Xianyang, and Weinan, encompassing multiple counties within each city. Field surveys employed a “W”-shaped five-point sampling method, as described by Ren et al. [12]. At each sampling point, the following metrics were recorded: (1) infestation rate: percentage of damaged plants per 100 sampled plants, calculated as $Infestation\;rate\;(\%)=\left(\frac{Number\;of\;damaged\;plants}{Total\;plants\;sampled}\right)\times 100$; (2) larval density: mean number of larvae per 100 plants, expressed as $Larvae\;per\;100\;plants=\left(\frac{Total\;larvae\;counted}{Total\;plants\;sampled}\right)\times 100$; (3) damage severity: rated on a 0–9 scale (0 = no damage; 9 = complete plant destruction) based on leaf defoliation and heartbreak feeding symptoms [8]. Data for 2019–2021 were provided by the Plant Protection Station of Shaanxi Province Agriculture and Rural Affairs Department, while 2022–2023 data were jointly collected by our research team and local plant protection.

### 2.2. Simulation of FAW Invasion Routes in Shaanxi

To simulate the invasion pathways of FAW in Shaanxi, we utilized pest occurrence data from 2019 to 2023, sourced from the Shaanxi Provincial Plant Protection Station, combined with tour field surveys. Adult invasion times were estimated based on observed larval instar, local daily mean temperatures, and FAW effective accumulated temperature. Meanwhile, adult emergence timing was based on the observed larval instar, local daily mean temperatures, and the effective accumulated temperature required for FAW development. Specifically, we used the third-instar larvae as the basis for back-calculating the timing of adult emergence. We adopted a threshold temperature of 11.3 °C and a degree-day constant (thermal constant) of 344.5 day·degrees for the estimation, which were determined based on previous research [9,13].

To simulate FAW migration pathways, HYSPLIT [14] runs were initialized at 1500 m above ground level (AGL). This altitude accounts for observed FAW flight behavior and topographic influences. Radar studies indicate FAW migrations occur in layered flows between 300–1500 m AGL, with concentration peaks varying by region and season [6,15]. Additionally, the Qinling Mountains (elevation up to 3767 m) likely force ascending airflows, as documented in other Noctuids (e.g., *Mythimna separata*) that adopt higher flight altitudes (1000–2000 m) to cross mountain barriers [16,17,18]. The simulation assumed two consecutive nights of flight, with the endpoint of the first night serving as the starting point for the second night. Trajectory data were processed in ArcGIS 10.7, where effective trajectories were selected based on height, direction, and source area pest status. These were then overlaid on provincial administrative maps obtained from the National Geomatics Center of China. By analyzing the simulated trajectories from 2019 to 2023, we identified potential source regions for FAW populations invading Shaanxi, providing crucial insights into their invasion dynamics.

### 2.3. Genetic Characterization of FAW Populations from Different Geographical Regions

A total of 113 FAW individuals from 12 different geographical populations were collected during 2022–2023, including multiple regions within Shaanxi Province and samples from Yunnan, Henan, and Hubei provinces (Table 1). DNA was extracted using QIAGEN DNA extraction kits following the manufacturer’s protocol. PCR amplification targeted the mitochondrial *COI* gene (F:TTCGAGCTGAATTAGGGACTC; R:GATGTAAAATATGCTCGTGT) and nuclear *Tpi* gene (F:GGTGAAATCTCCCCTGCTATG; R:AATTTTATTACCTGCTGTGG), using primers and conditions described by Zhang [19]. The *COI* gene was selected due to its high variability and maternal inheritance, making it suitable for identifying strain-specific maternal lineages. The *Tpi* gene, a nuclear marker, was chosen to provide insights into paternal lineages and potential hybridization events. This combination of markers allows for a comprehensive analysis of genetic diversity and population structure. In addition, we acknowledge that our phylogenetic analysis is not designed to definitively infer the geographic origins of FAW populations due to potential gaps in *COI* reference sequences from regions like Yunnan or Guangxi. Future research incorporating more comprehensive reference data could further clarify the geographic origins of FAW in Shaanxi.

PCR products were sequenced commercially, and sequences were analyzed using Geneious R11 software [20]. MEGA X software [21] was used for sequence alignment and construction of Neighbor-Joining phylogenetic trees with 1000 bootstrap replicates.

### 2.4. Preliminary Investigation of FAW Pupal Overwintering in Shaanxi

Laboratory-reared FAW pupae were used for the experiment. Thirty pupae of the same age were placed in 1500 mL plastic containers filled with 10 cm of soil, following the method of Li et al. [22]. On 10 December 2022, these containers were placed in fallow fields at six test sites across Shaanxi (Hantai District (106°55′ E, 33°09′ N) and Ningqiang (106°03′ E, 33°57′ N) County in Hanzhong, Xunyang (109°32′ E, 33°06′ N) County and Zhenping (109°26′ E, 31°51′ N) County in Ankang, and Danfeng (110°06′ E, 33°46′ N) County and Shanyang (110°16′ E, 33°27′ N) County in Shangluo. Pupal survival was checked every five days, following a nondestructive protocol. Pupae were initially identified within the topsoil layer (0–10 cm depth) without physical disturbance, with viability preliminarily assessed through intact morphology observation and gentle tactile response (e.g., leg twitching). Intact pupae were then carefully excavated using sterile tools, placed individually in ventilated containers with native soil substrate, and transferred to a climate-controlled chamber (25 ± 1 °C, 70 ± 5% RH, 12:12 L:D photoperiod) to monitor emergence. Mortality was confirmed if no movement or adult emergence occurred within 48 h of acclimatization at 20–22 °C. Three replicates (10 pupae per replicate) were processed per site, along with sterilized soil controls to account for environmental contamination. Data were statistically analyzed using SPSS software (Version 27.0) [23], with significant differences in pupal survival rates among sites were analyzed using one-way ANOVA followed by Tukey’s honestly significant difference (HSD) post hoc test (α = 0.05).

## 3. Results

### 3.1. Initial Occurrence of FAW in Shaanxi from 2019 to 2023

First detection dates showed significant interannual variation (ANOVA, *F* = 7.32, *p* < 0.01), progressing from 31 May (2019) to late June (2021–2023) (Table 2). In 2019, FAW was first detected on 31 May in Yangxian County, Hanzhong City, marking its initial invasion into Shaanxi Province with a relatively low infestation level. The first occurrence in 2020 was delayed to 28 June in Zhenping County, Ankang City, but with increased severity compared to 2019. From 2021 to 2023, initial detections consistently occurred between late June and early July, with a general declining trend in infestation severity. The specific survey data from different years are shown in the Supplement Files.

### 3.2. Simulation of FAW Migration Trajectories in Shaanxi

HYSPLIT model simulations of migration trajectories revealed that the main source regions for FAW invading Shaanxi from 2019 to 2023 included Sichuan, Chongqing, Hubei, Guizhou, and Henan (Figure 1). Three primary migration routes were identified: (1) the Sichuan route, mainly invading the Hanzhong area; (2) the Hubei-Chongqing route, primarily affecting the Ankang area; and (3) the Henan-Hubei route, mainly invading the Shangluo area. Notably, migration patterns varied across years. All three routes coexisted in 2019–2020; the Henan-Hubei and Sichuan routes predominated in 2021; and, in 2022–2023, an additional intra-provincial migration route emerged alongside the existing pathways. These variations may be attributed to inter-annual differences in meteorological conditions and source population sizes. For a detailed visualization of these trajectories, please refer to the Appendix A, where the HYSPLIT trajectory analysis results are provided. The suggested migration routes are considered tentative due to the lack of empirical data confirming the actual migration routes.

### 3.3. Molecular Characterization of FAW Biotypes from Different Geographical Populations

*COI* sequences revealed 85% rice-strain predominance, with corn-strain variants concentrated in northern Shaanxi (*χ*^2^ = 6.54, *p* = 0.01). Tpi gene analysis showed unexpected monomorphism across all populations (Table 3). Phylogenetic analysis showed that *COI*-strain 1 (rice strain) clustered with rice strains from Yunnan, India, South Africa, and the United States, while *COI*-strain 2 (corn strain) grouped with corn strains from the United States and Jiangsu (Figure 2). These results illuminate the genetic diversity and potential origins of FAW populations in Shaanxi.

### 3.4. Preliminary Investigation of FAW Overwintering in Shaanxi

To evaluate the overwintering potential of FAW in Shaanxi, pupal survival was monitored across six locations from 15 December 2022 to 14 January 2023. One-way ANOVA revealed significant differences in survival rates among locations. Post-hoc pairwise comparisons using Tukey’s HSD test confirmed that pupae failed to emerge at all sites, with survival rates declining progressively until complete mortality was observed by 14 January 2023 (Table 4 and Figure 3). No significant pairwise differences were detected between specific locations (all *p* > 0.05, Tukey’s HSD), suggesting uniformly lethal effects of low temperatures during the experimental period. These results indicate that Shaanxi’s winter conditions are unlikely to support FAW pupal overwintering or year-round breeding. However, it should be noted that the absence of continuous soil temperature data limits the explanatory power of this experiment regarding pupal mortality. Future studies would benefit from incorporating environmental data, such as soil temperature, to better understand the factors influencing FAW pupal survival.

## 4. Discussion

The FAW invasion in China has rapidly spread since 2019, posing a significant threat to agricultural production, particularly corn crops [24,25,26]. Our study reveals that FAW infestation dynamics in Shaanxi varied annually from 2019 to 2023, with a trend of delayed occurrence each year. The first detection in Shaanxi Province occurred on 31 May 2019, approximately 20 days later than in Hubei Province at a similar latitude [27]. Understanding FAW migration patterns and source regions is crucial for effective prediction and control [28]. Our study used the HYSPLIT platform to simulate FAW migration trajectories, a method widely applied in insect migration studies [29,30]. The results indicate that FAW migration routes and source regions invading Shaanxi varied annually from 2019 to 2023. Our findings have several direct implications for pest management strategies in Shaanxi and adjacent regions. The consistent genetic composition dominated by the rice strain suggests that monitoring and control efforts should focus on this strain. Additionally, the overwintering experiment results highlight that Shaanxi’s climate acts as a natural barrier to FAW’s year-round establishment. This implies that regional pest management could prioritize early-season monitoring and intervention, as FAW populations are likely to be re-invading each year rather than overwintering locally. Specifically, integrated pest management approaches could incorporate the timing of FAW migrations to optimize the deployment of control measures such as targeted insecticide applications or the release of natural enemies in alignment with the pest’s arrival. Furthermore, the stable strain composition observed could inform the selection of resistant maize varieties suited to counteract the predominant rice-strain FAW, thereby enhancing agricultural resilience against this invasive pest. Previous studies have identified two main migration routes for FAW in China, the western and eastern routes [10,11], and our findings largely support this pattern but also reveal more complex dynamics. For instance, we observed potential mixing of populations from eastern and western routes, particularly in Henan province. This complexity aligns with recent studies suggesting that while the two migration routes are generally distinct, some interchange occurs [9,31]. The migration patterns we observed in Shaanxi show similarities to those reported in neighboring regions. For example, Zhang et al. [32] found that FAW populations in Sichuan could migrate to Shaanxi, which our results corroborate. Similarly, our observation of FAW migrations from Hubei and Henan into Shaanxi aligns with the findings of Sun et al. [31] and Tan et al. [33], who reported complex migration patterns in central China. The genetic diversity and potential hybridization observed in our study have significant implications for FAW management. Understanding these genetic dynamics is crucial for developing effective pest management strategies, as emphasized by Westbrook et al. [6]. The high genetic diversity in Henan underscores its pivotal role in FAW population dynamics, suggesting it should be a focal point for intensive monitoring and targeted control efforts. These findings enhance our understanding of FAW population genetics and highlight the need for adaptive and region-specific management strategies.

Genetic analysis of FAW populations provides crucial insights into their complex migration patterns. Our comprehensive study, examining 113 FAW samples from 12 distinct geographical populations using *COI* and *Tpi* molecular markers, revealed a notable discrepancy between mitochondrial and nuclear markers. This finding aligns with previous research by Zhang et al. [19] and Wang et al. [34], suggesting a consistent pattern across different studies. Meanwhile, the research results also indicated that the FAW population in Shaanxi was predominantly the rice strain, with a certain proportion of the corn strain in northern Shaanxi. This finding is consistent with the results of Li et al. [9], which highlighted that the migration routes and population genetic structure of FAW in China are shaped by various factors, including geographic origin and migration pathways. According to Li et al. [9], the western migration route originates in Myanmar and Yunnan, passing through Guizhou, Chongqing, Sichuan, and Shaanxi before merging with the eastern route, while the eastern route starts from northern Thailand, Laos, Vietnam, Guangxi, and Guangdong, proceeding through southeastern and east-central provinces to ultimately reach the Huang-Huai-Hai and Northeast Regions. This suggests that FAW populations in Shaanxi may be influenced by both western and eastern routes, resulting in a mix of rice and corn strains. Further analysis found that FAW populations in Henan exhibited the highest genetic diversity, likely due to Henan’s position at the intersection of eastern and western migration routes, making it a hot spot for genetic exchange. This aligns with our observations of genetic mixing in Shaanxi’s FAW populations, indicating some degree of gene flow among populations from different regions. While our results suggest that Shaanxi might serve as a mixing zone for FAW populations, we caution that these interpretations are preliminary and would be strengthened by the inclusion of additional *COI* reference sequences from underrepresented regions such as Yunnan or Guangxi. Without these data, our analysis provides valuable insights, but cannot conclusively determine geographic origins.

The uniform results obtained from the *Tpi* gene analysis suggest a high level of genetic similarity across the sampled FAW populations. This finding may imply a recent population expansion or a bottleneck event that has reduced genetic diversity. It also has important implications for pest management, as higher genetic uniformity may indicate a more cohesive population that could respond similarly to control measures. However, this interpretation is limited by the scope of our sampling, which primarily focused on Shaanxi and the surrounding regions. Future studies should expand the sampling to include a broader geographic range to better assess the overall genetic diversity and structure of FAW populations in China.

The results of our overwintering experiment indicate that FAW pupae could not successfully overwinter at any of the test sites in Shaanxi. This finding is consistent with previous research, which has established that FAW lacks diapause capability, rendering it unable to endure extended periods of cold [10]. However, recent observations have reported the persistence of FAW larvae well into the winter months of November and December in Hubei and Henan provinces [34,35]. This discrepancy between our experimental results and field observations raises intriguing questions about the potential for rapid adaptation in FAW populations or the existence of microhabitats that might offer refuge from harsh winter conditions. It also underscores the complexity of FAW’s adaptive potential and highlights a critical gap in our understanding of its overwintering capabilities, particularly in transitional climate zones. It should be noted that our study did not directly measure soil temperature at the overwintering sites, which could influence the interpretation of pupal survival. Historical soil temperature data would provide valuable additional context to our conclusions on pupal mortality. We recommend that future research consider incorporating such environmental variables to further explore the factors influencing FAW pupal survival in different regions. In addition, our study’s sampling scope, while extensive within Shaanxi, does have limitations that should be acknowledged. The sampled regions may not fully represent the entire range of FAW genetic diversity in China, particularly given the pest’s known ability to migrate over long distances. This limitation underscores the need for collaborative research efforts across multiple regions to capture a more complete picture of FAW population dynamics.

## 5. Conclusions

Our comprehensive study on the invasion dynamics, migration patterns, genetic diversity, and overwintering capacity of FAW in Shaanxi Province from 2019 to 2023 provides critical insights for developing effective pest management strategies. The observed interannual variations in FAW occurrence and the identified migration routes highlight the complex nature of its invasion dynamics. The genetic diversity and strain composition of FAW populations in Shaanxi, characterized by a predominance of the rice strain, underscore the need for region-specific management approaches. The inability of FAW pupae to survive the winter in Shaanxi suggests that climatic conditions act as a natural barrier to its year-round establishment. This finding is particularly valuable for designing early warning systems and targeted control measures, emphasizing the importance of integrating ecological, genetic, and climatic data in pest management frameworks. However, the questions raised by the discrepancies in overwintering capabilities and the implications of the observed genetic uniformity highlight areas where further research is needed. Continued investigation into these aspects will be crucial for developing effective and sustainable management strategies for this invasive pest.

## Figures and Tables

**Figure 1 insects-16-00620-f001:**
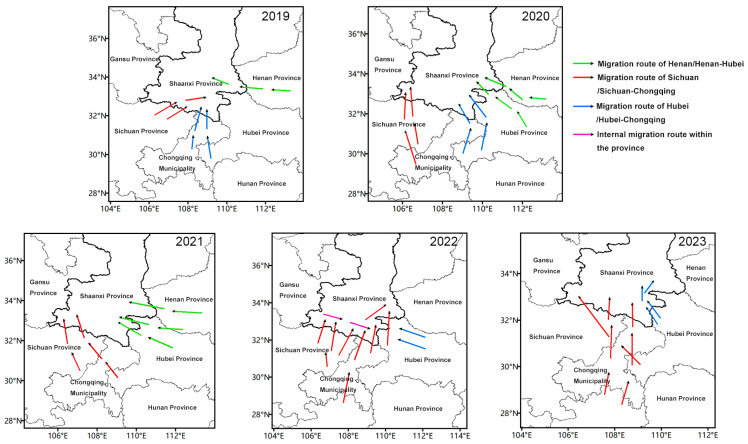
Invasion paths of FAW into Shaanxi Province from different migration routes in 2019–2023.

**Figure 2 insects-16-00620-f002:**
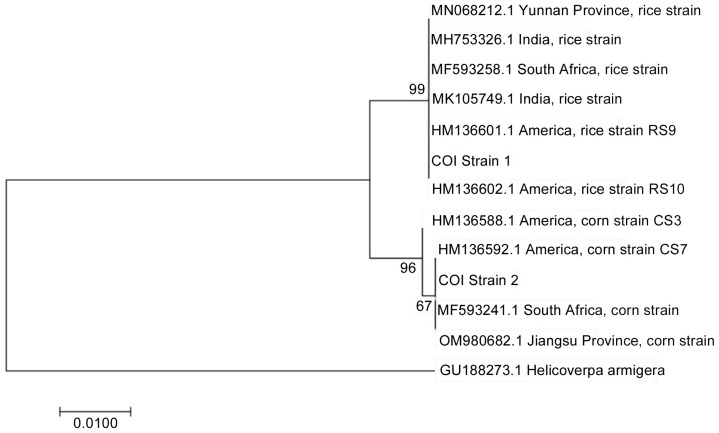
Phylogenetic tree inferred from *COI* gene fragments.

**Figure 3 insects-16-00620-f003:**
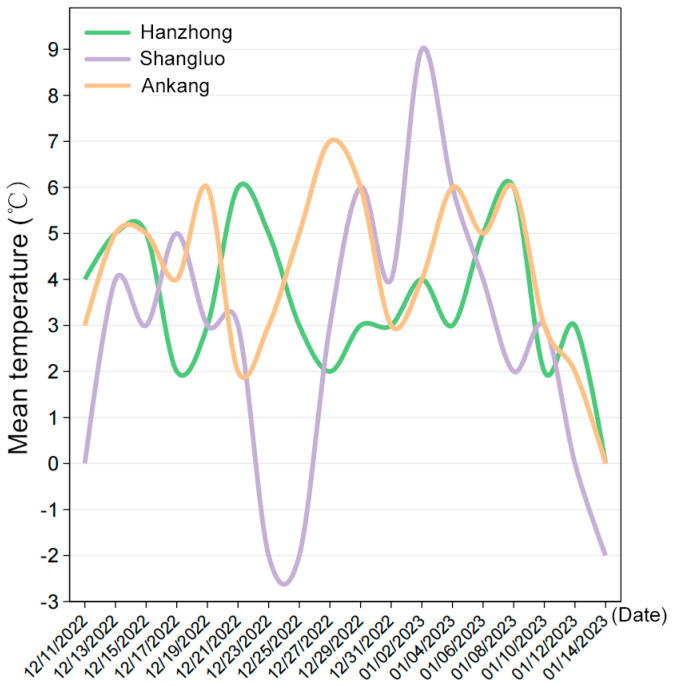
Daily average temperature in Hanzhong, Ankang, and Shangluo, respectively, in the winter of 2022–2023.

**Table 1 insects-16-00620-t001:** Sampling information for different geographical populations of FAW.

Sample No.	Collection	Collection Time	Host	Stage	Number of Samples
1	Baoji, Shaanxi Province (107°96′ E, 34°48′ N)	10/12/2023	Corn	larvae	13
2	Hanzhong, Shaanxi Province (107°12′ E, 34°48′ N)	7/20/2023	Corn	larvae	13
3	Ankang, Shaanxi Province (109°86′ E, 36°66′ N)	7/23/2023	Corn	larvae	7
4	Ankang, Shaanxi Province (109°26′ E, 32°52′ N)	8/28/2023	Corn	larvae	6
5	Ruili, Yunnan Province (97°49′ E, 23°58′ N)	4/25/2023	Corn	larvae	8
6	Xinxiang, Henan Province (114°08′ E, 32°10′ N)	7/5/2023	Corn	larvae	12
7	Shangluo, Shaanxi Province (110°21′ E, 34°01′ N)	8/3/2023	Corn	larvae	13
8	Wuhan, Hubei Province (114°06′ E, 30°34′ N)	6/30/2023	Corn	larvae	11
9	Xi’an, Shaanxi Province (109°08′ E, 34°25′ N)	9/7/2022	Corn	larvae	10
10	Hanzhong, Shaanxi Province (106°55′ E, 32°09′ N)	7/21/2022	Corn	larvae	7
11	Shangluo, Shaanxi Province (110°30′ E, 33°59′ N)	8/1/2022	Corn	larvae	6
12	Ankang, Shaanxi Province (108°36′ E, 32°50′ N)	7/26/2022	Corn	larvae	7

**Table 2 insects-16-00620-t002:** Summary of the earliest locations in Shaanxi and the damages of FAW in different years.

Survey Year	County	Initial Occurrence Date	Occurrence Area (mu)	Average Rate of Damaged Plants (%)	Average Larval Number per 100 Plants	Larval Instar	Estimated Immigration Date
2019	Yangxian (107°44′ E, 33°07′ N)	31 May	0.37	7.1	3	2–4	21 May
2020	Zhenping (109°26′ E, 31°51′ N)	28 June	40	7	15	1–4	15 June
2021	Chenggu (107°11′ E, 33°27′ N)	22 June	7	2.6	4	3–5	8 June
2022	Ningqiang (106°03′ E, 33°57′ N)	11 July	3.7	2.29	2	2–4	28 June
2023	Mian (106°54′ E, 33°09′ N)	20 June	1.2	1.08	2	2–5	1 June

**Table 3 insects-16-00620-t003:** Strain identification of FAW based on different molecular markers.

Sample No.	Collect Locations	*COI* Maker		*Tpi* Maker	
		Corn Strain	Rice Strain	Corn Strain	Rice Strain
1	Baoji, Shaanxi Province	4	9	13	0
2	Hanzhong, Shaanxi Province	0	13	13	0
3	Ankang, Shaanxi Province	0	7	7	0
4	Xianyang, Shaanxi Province	2	4	6	0
5	Ruili, Yunnan Province	0	8	8	0
6	Xinxiang, Henan Province	7	5	12	0
7	Shangluo, Shaanxi Province	4	9	13	0
8	Wuhan, Hubei Province	0	11	11	0
9	Xi’an, Shaanxi Province	0	10	10	0
10	Hanzhong, Shaanxi Province *	0	7	7	0
11	Shangluo, Shaanxi Province *	0	6	6	0
12	Ankang, Shaanxi Province *	0	7	7	0

* indicate that the larvae samples collected in 2022, other unmarked larvae samples collected in 2023.

**Table 4 insects-16-00620-t004:** Survival rates of pupae of FAW at different dates in Shaanxi (from December 2022 to January 2023).

Collect Locations	Survival Rates (%) of Pupae at Different Dates						
	15 December	20 December	25 December	30 December	4 January	9 January	14 January
Hanzhong	100 ± 0.00 a	95.55 ± 1.11 a	83.33 ± 1.92 ab	61.11 ± 2.94 ab	36.67 ± 1.93 ab	16.67 ± 1.93 a	0
Ankang	98.89 ± 1.11 a	96.67 ± 1.93 a	87.78 ± 1.11 a	64.59 ± 1.04 ab	37.78 ± 1.11 ab	6.67 ± 3.84 a	0
Shngluo	98.89 ± 1.11 a	88.89 ± 1.11 b	72.22 ± 1.11 c	55.55 ± 4.01 b	31.11 ± 2.22 ab	7.78 ± 1.14 a	0

Note: Data in the table are means ± SE. Means followed by different letters are significantly different (Tukey’s HSD test).

## Data Availability

The original contributions presented in this study are included in the article/Appendix A. Further inquiries can be directed to the corresponding authors.

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
