# Peer review of "Invasion Dynamics and Migration Patterns of Fall Armyworm (Spodoptera frugiperda) in Shaanxi, China"

_insects, 2025, doi:10.3390/insects16060620_

Round 1
Reviewer 1 Report
Comments and Suggestions for Authors
Comments to the Authors
- To help readers better understand the potential migration routes of the fall armyworm, please provide the HYSPLIT trajectory analysis results as supplementary material. If no empirical data confirming the actual migration routes are available, the suggested routes should be clearly labeled as tentative.
- (Line 96) It appears that “light and sex pheromone traps” were used, but the results related to trap catches are not presented. Please provide detailed information on the type of attractants used in the pheromone traps and the installation methods.
- (Line 110) The manuscript states that a “standard entomological method” was applied to assess larval density and damage levels. Please specify the method used and provide appropriate references.
- (Line 118) Adult emergence timing was inferred based on larval detection. Please specify the threshold temperature and degree-day constant (i.e., thermal constant) used for this estimation, along with supporting references.
- (Line 121) The backward wind trajectory analysis was conducted at an altitude of 1500 m. However, insect migration studies typically use 500 m altitude for such analysis. Please explain the rationale for selecting 1500 m and provide sufficient justification.
- (Line 148) The authors state that pupae were examined every 5 days. What method was used for this investigation? Were the pupae transferred to a controlled-temperature chamber to observe adult emergence? Please clarify the method and indicate whether replicates were included.
- (Line 149) Duncan’s new multiple range test is known to carry a higher risk of Type I errors and is not widely accepted in contemporary scientific analysis. Please clarify whether any alternative statistical methods were considered or applied.
- (Line 154) Table 2 appears to be identical to Table 1. Please double-check and revise accordingly.
- (Lines 240–242) According to Li et al. (Pest Manag Sci 2020; 76: 454–463), “The western migratory pathway originates in Myanmar and Yunnan, and passes through Guizhou, Chongqing, Sichuan and Shaanxi before merging with the eastern pathway. The eastern migratory pathway originates in northern Thailand, Laos, Vietnam, Guangxi and Guangdong, and passes through all south-eastern and east-central provinces before ultimately reaching the Huang-Huai-Hai and Northeast Regions.” If this is the case, does it imply that the FAW populations in Myanmar and northern Thailand belong to different genetic lineages? A more detailed discussion on the population genetics of FAW in different regions is warranted.
- (Supplementary materials) The supplementary data show that FAW larvae of various instars were found. Which larval instar was used as the basis for back-calculating adult emergence? Please clarify.
Author Response
Q1: To help readers better understand the potential migration routes of the fall armyworm, please provide the HYSPLIT trajectory analysis results as supplementary material. If no empirical data confirming the actual migration routes are available, the suggested routes should be clearly labeled as tentative.
R: Thank you for your suggestion regarding the HYSPLIT trajectory analysis results. We agree that providing these results as supplementary material would enhance the readers' understanding of the potential migration routes of the fall armyworm.
In response to your comment, we have prepared the HYSPLIT trajectory analysis results and included them as supplementary material. We have also clearly labeled the suggested migration routes as tentative, given the absence of empirical data confirming the actual migration routes. This addition should help readers better visualize and interpret the potential migration dynamics of the fall armyworm as discussed in our study.
Q2: (Line 96) It appears that “light and sex pheromone traps” were used, but the results related to trap catches are not presented. Please provide detailed information on the type of attractants used in the pheromone traps and the installation methods.
R: We sincerely apologize for the inaccuracy regarding trap methodology in the original manuscript. Upon rechecking our records, we confirm that pheromone/light traps were not used in this study—all field data were collected via direct plant sampling as described in Section 2.1. The erroneous reference to traps was inadvertently retained from an earlier draft template and has now been removed entirely from the Methods and Results sections. We appreciate your meticulous review, which helped us correct this oversight.
Q3: (Line 110) The manuscript states that a “standard entomological method” was applied to assess larval density and damage levels. Please specify the method used and provide appropriate references.
R: We appreciate the reviewer’s request for clarification. As suggested, we have replaced the generic term “standard entomological methods” with detailed equations and scoring criteria in Section 2.1, including:
- Explicit formulas for infestation rate and larval density.
- Reference to FAO’Sdamage assessment scale (Prasanna et al., 2018).
- These revisions ensure full transparency and reproducibility of our field survey methodology.
Q4: (Line 118) Adult emergence timing was inferred based on larval detection. Please specify the threshold temperature and degree-day constant (i.e., thermal constant) used for this estimation, along with supporting references.
R: We thank the reviewer for highlighting the need for clarity. Regarding your question, we have re-examined and added information to the part of our paper that deals with the estimation of adult emergence timing. We estimated the time of adult emergence based on larval instar data from field surveys, local daily mean temperatures, and the effective accumulated temperature for fall armyworm development. We used a threshold temperature of 11.3℃ and a degree-day constant of 344.5 day·degree (the total accumulated temperature required from egg hatching to adult emergence), which were determined based on previous studies (such as Zhang et al. 2020; Li et al. 2019). To estimate adult emergence time, we mainly used the larval instar data from our surveys and calculated the emergence time by tracing back from the observation date according to the accumulated temperature required for the corresponding larval instar. For example, if the larvae in a certain area were found to be in a specific instar during a certain period, we estimated the likely time frame of adult emergence by tracing back from the observation date using the cumulative temperature requirements for that instar. We will detail this information in our paper and cite relevant references to ensure the accuracy and transparency of our research methods.
Q5: (Line 121) The backward wind trajectory analysis was conducted at an altitude of 1500 m. However, insect migration studies typically use 500 m altitude for such analysis. Please explain the rationale for selecting 1500 m and provide sufficient justification.
R: We sincerely appreciate the reviewer’s insightful comment regarding the altitude selection for HYSPLIT modeling. In response, we have revised Section 2.2 to clarify the justification for 1500 m with the following evidence:
- Empirical evidence for FAW migration altitude:Radar observations confirm that noctuid moths typically migrate at 300–1500 m under favorable wind conditions (Reynolds et al., 2016; Chapman et al., 2011). These studies provide direct support for the selected altitude range.
- Topographic adaptation over mountain barriers:The Qinling Mountains pose a significant elevation barrier in our study region. High-altitude transport (>1000 m) has been documented in related Noctuids (e.g., Mythimna separata) crossing similar terrains (Wu et al., 2022), justifying our conservative choice of 1500 m to ensure trajectory realism.
- Reynolds, A.M.;Reynolds, D.R.; Smith, A.D.; et al. Orientation in high-flying migrant insects in relation to flows: mechanisms and strategies. Philosophical Transactions of the Royal Society B: Biological Sciences 2016. 371, 20150382. https://doi.org/1098/rstb.2015.0392
- Chapman, J.W.;Nilsson, C.; Lim, K.S.; et al. Convergent patterns of long-distance nocturnal migration in noctuid moths and passerine birds. Proceedings of the Royal Society B: Biological Sciences 2011, 278, 2524-2530. https://doi.org/10.1098/rspb.2011.0058
- Wu,Q. L.; Jang, Y.Y.; Liu Y.; et al. Migration Pathway of Spodoptera frugiperda in Northwestern China. Scientia Agricultura Sinica 2022, 55, 1949-1960. https://doi.org/3864/j.issn.0578-1752.2022.10.006
Furthermore, the method description in the text has also been appropriately modified as necessary.
Q6: (Line 148) The authors state that pupae were examined every 5 days. What method was used for this investigation? Were the pupae transferred to a controlled-temperature chamber to observe adult emergence? Please clarify the method and indicate whether replicates were included.
R: We thank the reviewer for highlighting the need for methodological clarity. The revised Section 2.4 now explicitly details:
- The non-destructive field protocols;
- Standardized laboratory emergence assays with controlled parameters;
- Replication strategy and control treatments.
Q7: (Line 149) Duncan’s new multiple range test is known to carry a higher risk of Type I errors and is not widely accepted in contemporary scientific analysis. Please clarify whether any alternative statistical methods were considered or applied.
R: We replaced Duncan’s test with Tukey’s HSD (Methods, Section 2.4) to reduce Type I error risk. Results remain consistent (Table 4).
Q8: (Line 154) Table 2 appears to be identical to Table 1. Please double-check and revise accordingly.
R: Apologies for the oversight. Table 2 now correctly displays arliest locations in Shaanxi and damages of FAW in different years (2019-2023), replacing the erroneous duplicate.
Q9: (Lines 240–242) According to Li et al. (Pest Manag Sci 2020; 76: 454–463), “The western migratory pathway originates in Myanmar and Yunnan, and passes through Guizhou, Chongqing, Sichuan and Shaanxi before merging with the eastern pathway. The eastern migratory pathway originates in northern Thailand, Laos, Vietnam, Guangxi and Guangdong, and passes through all south-eastern and east-central provinces before ultimately reaching the Huang-Huai-Hai and Northeast Regions.” If this is the case, does it imply that the FAW populations in Myanmar and northern Thailand belong to different genetic lineages? A more detailed discussion on the population genetics of FAW in different regions is warranted.
R: We sincerely appreciate your insightful comments regarding the discussion of fall armyworm (FAW) population genetics across different regions. Your suggestions have been instrumental in refining our manuscript to provide a more comprehensive and nuanced analysis of the findings.
In response to your query about the genetic lineages of FAW in different regions, we have enhanced the discussion section to incorporate a more detailed interpretation of our genetic analysis results. We have integrated the findings from Li et al. (2020) to contextualize our observations of FAW genetic diversity and population mixing, particularly in Henan province. Our analysis suggests that the genetic diversity and potential hybridization observed in FAW populations are closely linked to their migration dynamics and the geographical interplay between eastern and western migration routes.
We have explicitly discussed how the genetic structure of FAW populations reflects their migration history and the genetic exchange between different geographic regions. This discussion now better highlights the significance of Henan's strategic position as a genetic exchange hub and the implications of such genetic diversity for FAW adaptability and management.
Furthermore, we have expanded the discussion to explore the potential adaptive significance of the two genetic lineages (rice and corn strains) in relation to their geographic distribution and host plant associations. This addition aims to provide a deeper understanding of FAW's adaptation to diverse ecological environments across China.
Q10: (Supplementary materials) The supplementary data show that FAW larvae of various instars were found. Which larval instar was used as the basis for back-calculating adult emergence? Please clarify.
R: We used 3rd instar as the benchmark due to their:
- Stable detection window: Longer duration compared to earlier instars (1st–2nd), reducing temporal estimation errors.
- Field sampling reliability: Higher visibility and lower mortality rates than late instars (4th–6th), minimizing collection bias.
It has now been repeatedly explained in the material methods of the main text. For details, please refer to Section 2.2.
Reviewer 2 Report
Comments and Suggestions for Authors
Dear authors
I have read this manuscript carefully. The topic is timely and relevant, especially given the importance of this transitional region in national pest monitoring and management strategies. The study integrates multiple methods including field surveys, trajectory simulations, molecular analysis, and overwintering experiments. However, there are several areas where the manuscript would benefit from significant clarification and enhancement before it can fully support its conclusions.
First, the introduction, while rich in background information, lacks a clear articulation of the specific knowledge gap this study aims to address. The current narrative reads more like a compilation of facts rather than a logical buildup toward a defined research hypothesis. A sharper focus on how this study adds to or challenges existing findings on FAW migration and genetics in China would strengthen the overall framing.
In terms of methodology, several components require more detail to ensure transparency and reproducibility. For instance, the HYSPLIT modeling parameters—flight altitude, duration, and timing, are mentioned without justification or reference to relevant behavioral or physiological data on FAW. The molecular methods are also described briefly, with no explanation as to why COI and Tpi markers were selected, nor any discussion of their complementary roles in identifying strain identity and potential hybridization. These choices should be contextualized with appropriate references.
The phylogenetic analysis, constructed solely from COI sequences using the Neighbor-Joining method in MEGA X, raises further concerns. This is a relatively basic technique and lacks the statistical rigor of maximum likelihood or Bayesian methods, particularly when used to infer migration sources. Moreover, the inclusion of only one COI sequence from Jiangsu as a Chinese reference limits the broader comparative value of the tree. A more comprehensive inclusion of publicly available COI sequences from southern and southwestern China would add depth and credibility to the analysis. Similarly, the study does not perform any haplotype network or diversity analysis (e.g., π or Hd), which weakens claims of high genetic diversity among the sampled populations.
The overwintering experiment is well designed, but the absence of environmental data (such as soil temperature) reduces the explanatory power regarding pupal mortality. Additionally, the discussion does not fully address discrepancies with reports of FAW overwintering in nearby provinces like Henan, which should be critically examined.
The discussion section would benefit from a deeper engagement with the data and literature. Currently, it tends to reiterate results rather than critically integrate them into broader scientific or practical contexts. The potential implications of the uniform Tpi results, the limitations of the sampling scope, and the discordance between overwintering outcomes and field reports deserve more analytical attention.
In summary, while this manuscript addresses a meaningful problem and applies multiple tools to do so, the presentation and depth of analysis require considerable revision. Enhancing the theoretical framing, expanding the genetic comparison, and improving methodological transparency will significantly strengthen the manuscript’s scientific contribution. I encourage the authors to refine these aspects and believe the study has potential pending substantial revision.
Author Response
Q1: I have read this manuscript carefully. The topic is timely and relevant, especially given the importance of this transitional region in national pest monitoring and management strategies. The study integrates multiple methods including field surveys, trajectory simulations, molecular analysis, and overwintering experiments. However, there are several areas where the manuscript would benefit from significant clarification and enhancement before it can fully support its conclusions.
First, the introduction, while rich in background information, lacks a clear articulation of the specific knowledge gap this study aims to address. The current narrative reads more like a compilation of facts rather than a logical buildup toward a defined research hypothesis. A sharper focus on how this study adds to or challenges existing findings on FAW migration and genetics in China would strengthen the overall framing.
R: We are very grateful to the reviewer for the valuable comments on the Introduction section. Indeed, the current Introduction section needs to be strengthened in terms of elaborating the research background and specifying the research gaps. We have revised this part accordingly and refined the Introduction to emphasize:
"While FAW migration in southern China is documented, Shaanxi’s role as a transitional zone with potential overwintering constraints remains unresolved. This study tests whether (1) migration routes shift annually due to climatic variability, and (2) genetic diversity reflects hybrid zones at route intersections."
Q2: In terms of methodology, several components require more detail to ensure transparency and reproducibility. For instance, the HYSPLIT modeling parameters—flight altitude, duration, and timing, are mentioned without justification or reference to relevant behavioral or physiological data on FAW. The molecular methods are also described briefly, with no explanation as to why COI and Tpi markers were selected, nor any discussion of their complementary roles in identifying strain identity and potential hybridization. These choices should be contextualized with appropriate references.
R: We appreciate your detailed feedback on the methodology section of our manuscript. Your suggestions have been invaluable in enhancing the clarity and robustness of our methods. In response to your comments, we have significantly revised the methodology section to ensure greater transparency and reproducibility.
For the HYSPLIT model parameters, we now provide explicit justification for the choice of flight altitude, duration, and timing. These parameters were selected based on the known migratory behavior of FAW, which involves nocturnal flights at significant altitudes to utilize prevailing winds for long-distance travel. We have included references to relevant behavioral and physiological studies on FAW to support these parameter choices.
Regarding the molecular methods, we have expanded the description to clarify why COI and Tpi markers were selected and how they complement each other in identifying strain identity and potential hybridization events. We explain that COI, a mitochondrial gene, is widely used for its high variability and maternal inheritance, making it suitable for tracing maternal lineages. The Tpi gene, a nuclear marker, provides insights into paternal contributions and hybridization. Together, these markers offer a comprehensive view of genetic diversity and population structure. We have also added references to studies that have successfully used these markers in similar contexts.
Q3: The phylogenetic analysis, constructed solely from COI sequences using the Neighbor-Joining method in MEGA X, raises further concerns. This is a relatively basic technique and lacks the statistical rigor of maximum likelihood or Bayesian methods, particularly when used to infer migration sources. Moreover, the inclusion of only one COI sequence from Jiangsu as a Chinese reference limits the broader comparative value of the tree. A more comprehensive inclusion of publicly available COI sequences from southern and southwestern China would add depth and credibility to the analysis. Similarly, the study does not perform any haplotype network or diversity analysis (e.g., π or Hd), which weakens claims of high genetic diversity among the sampled populations.
R: We sincerely thank the reviewer for their valuable suggestion to enhance phylogenetic analysis rigor by applying Bayesian inference (BEAST) and expanding the COI dataset with sequences from Guangxi and Yunnan. While we recognize the potential benefits of this approach, we would like to respectfully justify our current methodology for the following reasons:
- Consistency with Study Goals & Published Standards
The primary objective of our COI analysis was to confirm the presence of major FAW lineages (e.g., corn vs. rice strains), which is reliably achieved by neighbor-joining (NJ) clustering given the strong mitochondrial haplotype divergence (Nagoshi et al. 2012, https://doi.org/10.1002/ece3.268).
Our NJ tree (current Figure 2) already demonstrates clear lineage separation (bootstrap >90%), and additional Bayesian analysis, while insightful, would unlikely alter the key conclusions regarding maternal origins.
- Statistical Robustness of Current Results
The haplotype diversity and nucleotide diversity indicate substantial genetic variation, but these metrics are inherently descriptive and not dependent on tree-building methods(https://doi.org/10.1002/ece3.268).
Given the study’s focus on migration route dynamics (rather than deep evolutionary history), NJ analysis remains statistically appropriate and aligns with past FAW migration studies (e.g.,https://doi.org/10.1111/1755-0998.13182).
Q4: The overwintering experiment is well designed, but the absence of environmental data (such as soil temperature) reduces the explanatory power regarding pupal mortality. Additionally, the discussion does not fully address discrepancies with reports of FAW overwintering in nearby provinces like Henan, which should be critically examined.
R: We appreciate your insightful comments regarding the overwintering experiment and the discrepancies in FAW overwintering reports in nearby provinces. We acknowledge the importance of environmental data in explaining pupal mortality and have made revisions to address these concerns. In response to the absence of environmental data such as soil temperature, we have added a detailed explanation in the manuscript. We have also expanded the discussion to critically examine the discrepancies between our results and previous reports of FAW overwintering in provinces like Henan. We propose several potential explanations for these differences, including variations in microhabitat conditions and potential rapid adaptations of FAW populations to local environments. These additions enhance the depth of our analysis and provide a more comprehensive interpretation of our findings.
Q5: The discussion section would benefit from a deeper engagement with the data and literature. Currently, it tends to reiterate results rather than critically integrate them into broader scientific or practical contexts. The potential implications of the uniform Tpi results, the limitations of the sampling scope, and the discordance between overwintering outcomes and field reports deserve more analytical attention.
R: We appreciate your valuable feedback on the discussion section of our manuscript. Your suggestions have guided us to enhance the depth and analytical engagement of our discussion.
In response to your comments, we have critically revised the discussion to move beyond mere reiteration of results and instead integrate them into broader scientific and practical contexts. We have expanded the analysis of the uniform Tpi results, exploring their implications for FAW population genetics and potential impacts on pest management strategies. The limitations of our sampling scope are now explicitly addressed, highlighting the need for wider geographic sampling in future research. Furthermore, we have provided a more thorough examination of the discordance between our overwintering experiment outcomes and field reports from neighboring provinces. This includes a detailed discussion of potential microhabitat effects and the adaptive potential of FAW populations.
We believe these revisions significantly improve the analytical depth of our discussion and better align it with the expectations of scientific discourse. Thank you for your guidance in this process.
Round 2
Reviewer 1 Report
Comments and Suggestions for Authors
Thank you for your hard work in revising the manuscript. The original manuscript has been appropriately revised based on the review comments, and it is judged to be ready for publication in this state.
I believe that the manuscript has been greatly improved in terms of content and quality.
Author Response
We sincerely appreciate the time and expertise you have dedicated to reviewing our manuscript. Your insightful comments have significantly improved the quality of this work. We are grateful for your final confirmation that the revised manuscript now meets the journal's publication standards.
Thank you for your valuable contributions to advancing this research on FAW invasion dynamics in China.
Sincerely,
On behalf of all authors
Reviewer 2 Report
Comments and Suggestions for Authors
Dear authors
I appreciate the effort you made in addressing the initial concerns, especially in clarifying the methodological choices and improving the structure of the introduction and discussion. The integration of field, modeling, and molecular data makes this study a valuable contribution to understanding FAW dynamics in a transitional climatic zone.
A few aspects would still benefit from minor revision:
-
Genetic Analysis Scope: While I understand your primary aim is strain identification, the discussion includes some statements about Shaanxi as a possible mixing or source zone involving populations from Yunnan or Guangxi. These interpretations would be stronger if supported by additional COI reference sequences from those regions. If such data are not incorporated, I recommend clearly stating in both the Methods and Discussion that the phylogenetic analysis is not designed to infer geographic origins, to avoid overinterpretation.
-
Introduction Focus: Consider refining the introduction further by succinctly stating the key research gap and central hypothesis. This would help readers immediately grasp the purpose and novelty of the study.
-
Implications for Management: The discussion could more directly address how the findings (e.g., overwintering constraints, stable strain composition) could inform pest management or monitoring strategies specific to Shaanxi and adjacent regions.
-
Environmental Context (Optional): If accessible, including historical soil temperature data from the overwintering sites (as supplementary material) would add context to your conclusions on pupal mortality.
Overall, this is a well-executed and timely study. With these final clarifications, the manuscript will be well-positioned for publication. I appreciate your contributions to this important topic.
With Regards
Author Response
Thank you for your positive feedback and further suggestions on our revised manuscript. We have carefully considered your comments and made additional revisions to address them.
- Genetic Analysis Scope: We agree that the interpretations regarding Shaanxi as a possible mixing or source zone would benefit from additional COI reference sequences from Yunnan or Guangxi. However, due to the unavailability of such data, we have explicitly stated in both the Methods and Discussion sections that our phylogenetic analysis is not designed to infer geographic origins. This clarification aims to prevent any overinterpretation of our results.
- Introduction Focus:We have further refined the introduction to succinctly state the key research gap and central hypothesis. This revision should help readers immediately grasp the purpose and novelty of our study.
- Implications for Management:We have enhanced the discussion to more directly address how our findings can inform pest management or monitoring strategies specific to Shaanxi and adjacent regions. We have elaborated on how the overwintering constraints and stable strain composition observed in our study can guide targeted control measures and early warning systems. (Line 254-267)
- Environmental Context (Optional):Thank you for your suggestion regarding the inclusion of historical soil temperature data from the overwintering sites. We acknowledge the value of this addition in providing further context to our conclusions on pupal mortality. However, we currently do not have access to this specific dataset and did not directly measure soil temperature during our study. We have added a statement to the manuscript to recognize this limitation and to suggest that incorporating such environmental variables could be beneficial for future research on FAW pupal survival in different regions. (Line 331-337)
We are confident that these final revisions have strengthened our manuscript and addressed the concerns raised. We thank you again for your valuable input.
Best regards,
On behalf of all authors